# Convergence of No-Swap-Regret Dynamics in Self-Play

**Renato Paes Leme**
Google Research

**Georgios Piliouras**
Google Deepmind

**Jon Schneider**
Google Research

## Abstract

In this paper, we investigate the question of whether no-swap-regret dynamics have stronger convergence properties in repeated games than regular no-external-regret dynamics. We prove that in almost all *symmetric* zero-sum games under *symmetric* initializations of the agents, *no-swap-regret* dynamics in self-play are guaranteed to converge in a strong "frequent-iterate" sense to the Nash equilibrium: in all but a vanishing fraction of the rounds, the players must play a strategy profile close to a symmetric Nash equilibrium. Remarkably, relaxing any of these three constraints, i.e. by allowing either i) asymmetric initial conditions, or ii) an asymmetric game or iii) no-external regret dynamics suffices to destroy this result and lead to complex non-equilibrating or even chaotic behavior.

In a dual type of result, we show that the power of no-swap-regret dynamics comes at a cost of imposing a time-asymmetry on its inputs. While no-external-regret dynamics can be completely determined by the cumulative reward vector received by each player, we show there does not exist any general no-swap-regret dynamics defined on the same state space. In fact, we prove that any no-swap-regret learning algorithm must play a time-asymmetric function over the set of previously observed rewards, ruling out any dynamics based on a symmetric function of the current set of rewards.

## 1 Introduction

The analysis of learning dynamics in games is a well-established problem situated at the intersection of game theory, online optimization and evolutionary game theory [16, 47, 54]. The significance of this area has been amplified by the emergence of prominent machine learning architectures and applications relying on multi-agent, typically zero-sum, games [42, 32, 46, 52, 14, 49, 10]. Symmetric zero-sum games and their dynamics are actually of particular interest both from a traditional evolutionary perspective [54, 13] as well as from a modern Machine Learning perspective as creating a population of agents that compete against each other in a heads-up fashion to outperform each other ("self-play") has been shown to be a reliable recipe for creating super-humanly capable agents for a wide range of tasks [9, 38].

Despite growing interest in understanding and predicting the long-term behavior of such systems, recent studies have revealed a wide array of negative results, demonstrating the elusiveness of game dynamics. These range from non-convergence results to the establishment of chaotic or even essentially arbitrary behavior [44, 28, 30, 40, 3, 4, 55, 26, 45]. Notably, these instability and chaotic behaviors persist even in the analysis of well-known online optimization algorithms, such as Multiplicative Weights Updates (MWU/Hedge), Online Gradient Descent, Follow-the-Regularized-Leader a.o., even within the narrow but seminal class of (symmetric) zero-sum games [7, 18, 19, 20].

Although it is possible to stabilize learning dynamics in zero-sum games using e.g., optimistic variants of MWU [22, 43, 23], such results leave something to be desired as they presuppose that the agents coordinate to use a specific instantiation of a learning algorithm. Ideally, we would like instead to be able to prove such strong convergence results based on more abstract properties of the dynamics.

38th Conference on Neural Information Processing Systems (NeurIPS 2024).

Swap (internal) regret is once such abstract property of online learning dynamics. Unlike the more permissive case of no-(external) regret, where the algorithm's performance needs to compete against the best fixed action with hindsight, no-swap-regret algorithms need to compete against the best adaptive deviation policy with hindsight (i.e., for each occurrence of action $i$ we consider to the best possible deviating action $j$ with hindsight). Somewhat surprisingly, it is possible to adapt no-regret algorithms to no-internal-regret algorithms efficiently [11, 53] with very recent work providing further efficient such reductions [21, 48]. The stronger nature of swap regret results into numerous applications, such as (multi)-calibration [31, 37, 51, 35, 33], robustness against dynamic strategic behavior [24, 41, 15, 5] and AI Safety [17]. Arguably, however, its most important application is due to its tight connection to correlated equilibria, introduced by Aumann [6], as it is well known that the time-average empirical distribution of play resulting from no-internal/swap regret algorithms is guaranteed to converge to the set of correlated equilibria (CE)[1], a (typically strict) relaxation of the predominant game theoretic solution concept of Nash equilibria. In contrast, no-regret algorithms only guarantee time-average convergence to the even more relaxed solution concept of coarse correlated equilibria (CCE) (see preliminary section for precise definitions). Interestingly, in the special case of zero-sum games, for almost all but pathological zero-measure instances of them, the notions of Nash equilibria and correlated equilibria coincide and are in fact unique (whereas CCE do not). This opens the following tantalizing possibility:

*Does no-swap-regret minimization suffice for Nash convergence in (almost) all zero-sum games?*

The answer to above question is strongly negative. Even in trivial two strategy zero-sum games, such as Matching Pennies, swap regret minimization does not suffice for convergence. Interestingly, however, a sweeping positive result holds for the case of **symmetric** zero-sum games.

**Informal theorem:** In almost all symmetric zero-sum games, under arbitrary symmetric initializations for both agents, any no-swap-regret algorithm in self-play is guaranteed to converge to the Nash equilibrium, except[2] for a vanishingly small fraction of iterates.

At a technical level, the result depends on two different arguments. First, even in the case of symmetric zero-sum games it is still possible to show that generically the correlated equilibria remain unique, however, the argument does not follow the analysis of general zero-sum games as symmetric cases are themselves non-generic within the larger class. Secondly, we leverage the symmetry of the trajectories to show that time-averaged convergence of symmetric action profiles to a product distribution implies the desired convergent behavior.

This unexpected connection between swap regret minimization and symmetry in games inspires the investigation of other ways that symmetry can insert itself in the study of online learning itself. For example, it is well understood that most standard no-regret algorithms such as MWU, can be completely determined by the vector of cumulative rewards and thus their outputs remain invariant to any permutation of their history, i.e. they exhibit a strong type of time-symmetry. Interestingly, we show that such time-symmetry is provably at odds with swap regret minimization. At a technical level, this argument is based on a construction that couples the behavior of online learning dynamics to particular classes of card guessing games (e.g., [25]) that enable precise control over the algorithm's optimal expected utility (in particular, showing that the play of any such algorithm must be very close to the Follow-The-Leader algorithm, at least when averaged over certain segments of time).

## 2 Model and Preliminaries

### 2.1 Games and learning

We consider a setting where two learners (Alice and Bob) are repeatedly playing a game $G$ for $T$ rounds. We assume the game $G$ has $N$ actions for both Alice and Bob, and Alice and Bob will play mixed strategies belonging to the $N$-dimensional simplex $\Delta_N$. The game $G$ can be thought of as a pair of bilinear functions $(G_A, G_B)$ describing the payoffs for Alice and Bob: in round $t$, if Alice plays action $a_t \in \Delta_N$ and Bob plays action $b_t \in \Delta_N$, then Alice receives payoff $G_A(a_t, b_t)$ and Bob receives utility $G_B(a_t, b_t)$. One specific class of games we consider are *zero-sum games*, where

---

[1]Due to their connections to equilibria, establishing fast swap regret minimization (and variations thereof) for different classes of games is a subject of a lot of recent work (e.g., [1, 2, 50, 56, 27]).

[2]This minor disclaimer is necessary as it is always possible to inject such vanishingly small "noise" in any trajectory without affecting its time-average regret.

$G_B(a_t, b_t) = -G_A(a_t, b_t)$; for such games we will omit the subscript and write $G(a_t, b_t)$ to denote $G_A(a_t, b_t)$.

For our purposes, a learning algorithm for Alice in this repeated game is a function which maps the history of played mixed strategies (e.g., $a_1, b_1, a_2, b_2, \ldots, a_{t-1}, b_{t-1}$) to the mixed strategy that Alice will play next ($a_t$). Learning algorithms for Bob are defined symmetrically. Note that we operate in the deterministic full-information setting where both Alice and Bob know the game $G$ and can see each other's mixed strategy after each round[3].

We consider two classes of learning algorithms, *no-regret algorithms* and *no-swap-regret algorithms*. Alice's (external) regret is defined via

$$\mathsf{Reg}_A = \sum_{t=1}^{T} G_A(a_t, b_t) - \max_{a^* \in \Delta_N} \sum_{t=1}^{T} G_A(a^*, b_t) \tag{1}$$

(with Bob's external regret $\mathsf{Reg}_B$ defined similarly). Alice's swap regret is defined via

$$\mathsf{SwapReg}_A = \sum_{t=1}^{T} G_A(a_t, b_t) - \max_{\pi_A : [N] \to [N]} \sum_{t=1}^{T} G_A(\pi_A(a_t), b_t). \tag{2}$$

In (2), the maximum is over all "swap functions" $\pi$ mapping the set of $N$ actions to itself. We extend $\pi$ to act on mixed strategies (elements of $\Delta_N$) in the natural way (i.e., $\pi(x)_i = \sum_j x_j \cdot \mathbf{1}(\pi(j) = i)$). We say Alice's learning algorithm is *no-regret* if it is guaranteed that $\mathsf{Reg}_A = o(T)$. Similarly, we say it is *no-swap-regret* if it is guaranteed that $\mathsf{SwapReg}_A = o(T)$. It is known that both efficient no-regret and no-swap-regret algorithms exist, with regret scaling as $\widetilde{O}(\sqrt{T})$ [11] .

### 2.1.1 Symmetric games and symmetric learners

In this note we primarily consider symmetric learning dynamics in symmetric games. A game $G$ is *symmetric* if $G_A(a_t, b_t) = G_B(b_t, a_t)$. As with zero-sum games, for symmetric games we will omit subscripts and use $G(a_t, b_t)$ to refer to $G_A(a_t, b_t)$. Some games are both symmetric and zero-sum (e.g., the Rock-Paper-Scissors example we introduce later in Example 1).

In a symmetric game, it's natural to consider the setting where Alice and Bob play identical learning algorithms with identical initialization. This results in completely symmetric learning dynamics for Alice and Bob (i.e., $a_t = b_t$ for all rounds $t$). We write $x_t \in \Delta_N$ to denote the common strategy that Alice and Bob play at time $t$.

### 2.2 Equilibria in games

We are interested in the convergence of various types of learning algorithms to specific equilibria of $G$. We begin by defining the equilibria of interest.

For a game $G$, a (not necessarily symmetric) *joint strategy profile* $\sigma$ is a distribution over all $N^2$ pairs of pure strategies for Alice and Bob. Note that this is not necessarily a product distribution, and in particular allows for Alice's mixed strategy and Bob's mixed strategy to be correlated. It is convenient to identify the set of joint distributions $\Delta_{N^2}$ with the convex subset $\mathcal{S}$ of the tensor product space $\mathbb{R}^N \otimes \mathbb{R}^N$ defined as the convex hull of all elements of the form $a \otimes b$ where $a, b \in \Delta_N$. In particular, given $a, b \in \Delta_N$, the element $a \otimes b$ corresponds to the joint strategy profile where Alice plays mixed strategy $a$ and Bob independently plays mixed strategy $b$. In other words, the pair $(i, j)$ is played with probability $a_i b_j$.

We consider three different types of equilibria, which are (in increasing order of fineness) coarse-correlated equilibria, correlated equilibria, and Nash equilibria. We define these below:

- A *coarse-correlated equilibrium* is a joint strategy profile where neither Alice nor Bob has an incentive to unilaterally deviate to a single pure action. Formally, $\sigma$ is a coarse-correlated equilibrium if both

---

[3]Alternatively, everything we describe also holds in a slightly weaker setting, where the players do not know the game $G$ but instead after each round each player sees the counterfactual payoffs they would have received for each of their possible $N$ actions.

$$\mathbb{E}_{(i,j)\sim\sigma}[G_A(i,j)] \geq \mathbb{E}_{(i,j)\sim\sigma}[G_A(i^*,j)], \forall i^* \in [N]$$

$$\mathbb{E}_{(i,j)\sim\sigma}[G_B(i,j)] \geq \mathbb{E}_{(i,j)\sim\sigma}[G_B(i,j^*)], \forall j^* \in [N]$$

- A *correlated equilibrium* is a joint strategy profile where neither Alice nor Bob has an incentive to deviate from their assigned action, where their deviation may depend on their original action. Formally, $\sigma$ is a correlated equilibrium if both

$$\mathbb{E}_{(i,j)\sim\sigma}[G_A(i,j)] \geq \mathbb{E}_{(i,j)\sim\sigma}[G_A(\pi_A(i),j)], \forall \pi_A : [N] \rightarrow [N]$$

$$\mathbb{E}_{(i,j)\sim\sigma}[G_B(i,j)] \geq \mathbb{E}_{(i,j)\sim\sigma}[G_B(i,\pi_B(j))], \forall \pi_B : [N] \rightarrow [N]$$

- Finally, $\sigma$ is a *Nash equilibrium* if $\sigma$ is a product distribution $\sigma = a \otimes b$ and neither Alice nor Bob has an incentive to deviate ($G_A(a,b) \geq G_A(a',b)$ and $G_B(a,b) \geq G_B(a,b')$). Note that we can alternatively think of Nash equilibria as the intersection of coarse-correlated (or correlated) equilibria with the set of product distributions.

**Example 1.** Consider the Rock-Paper-Scissors zero-sum game, defined via:

$$G(i,i) = 0, G(i,(i+1) \bmod 3) = 1, G(i,(i-1) \bmod 3) = -1.$$

The unique correlated equilibrium (and hence unique Nash equilibrium) in this game is the product distribution $\left(\frac{e_1+e_2+e_3}{3}\right) \otimes \left(\frac{e_1+e_2+e_3}{3}\right)$. However, the set of coarse correlated equilibria is much larger – it contains, for example, the element $\frac{1}{3}(e_1 \otimes e_1 + e_2 \otimes e_2 + e_3 \otimes e_3)$. Here there is no incentive to unilaterally deviate, but there is an incentive to deviate based on your choice of action (e.g., whenever you play $e_1$, you can improve your utility by instead playing $e_3$).

## 2.3 Convergence to equilibria

When players run learning algorithms in repeated games, they may over time converge to an equilibrium. There are three senses in which this may happen (that we discuss here). We say that Alice and Bob's strategies have *time-averaged convergence* to a joint strategy profile $\sigma$ if

$$\lim_{T\to\infty} \left\| \left(\frac{1}{T}\sum_{t=1}^{T} a_t \otimes b_t\right) - \sigma \right\| = 0.$$

In other words, time-averaged convergence means that the average of the joint strategy profiles Alice and Bob play converges over time to $\sigma$.

Likewise, we say that Alice and Bob's strategies have *frequent-iterate convergence* to $\sigma$ if for any $\varepsilon > 0$,

$$\lim_{T\to\infty} \Pr_{t\leq T}\left[\|(a_t \otimes b_t) - \sigma\| > \varepsilon\right] = 0.$$

Here the probability is taken over $t$ being drawn uniformly at random from all rounds between 1 and $T$. In other words, frequent-iterate convergence means that, as time goes on, almost all joint strategies profiles Alice and Bob play will be arbitrarily close to $\sigma$.

Finally, we say that Alice and Bob's strategies have *last-iterate convergence* to $\sigma$ if

$$\lim_{T\to\infty} \|(a_T \otimes b_T) - \sigma\| = 0.$$

Last-iterate convergence means that the sequence of joint action profiles played by Alice and Bob directly converge to $\sigma$. Note that last-iterate convergence is a stronger property than frequent-iterate convergence, which in turn is a stronger property than time-averaged convergence.

It is known that if Alice and Bob run certain types of learning algorithms, they will have time-averaged convergence to a certain type of equilibrium. Here are some known facts about learning dynamics []:

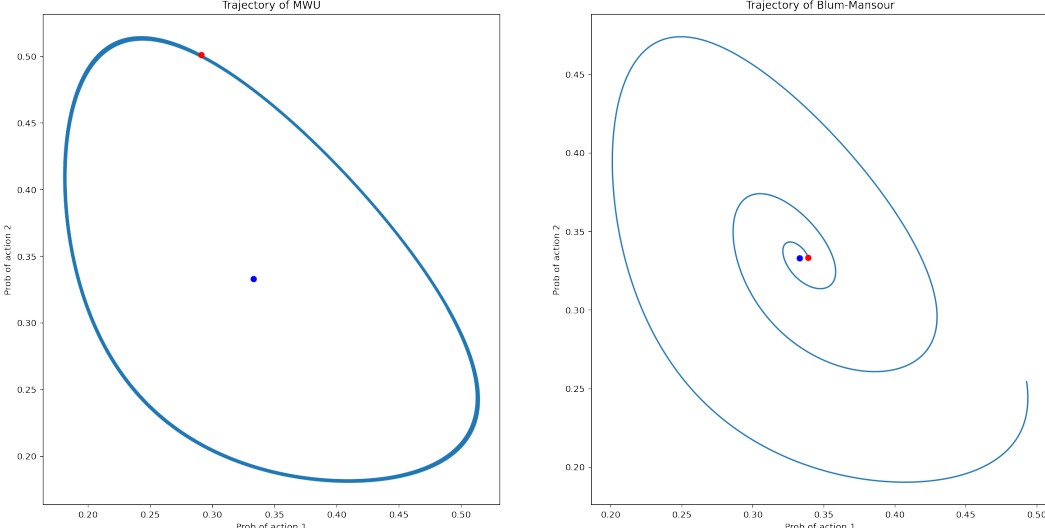

Figure 1: Trajectories of two learning algorithms (Multiplicative Weights and Blum-Mansour) playing Rock-Paper-Scissors. The axis corresponds to the probability of playing Rock and Paper. The blue dot corresponds to the probabilities at Nash equilibrium while the red dot is the last iterate after 10000 steps with learning rate $\eta = 0.001$.

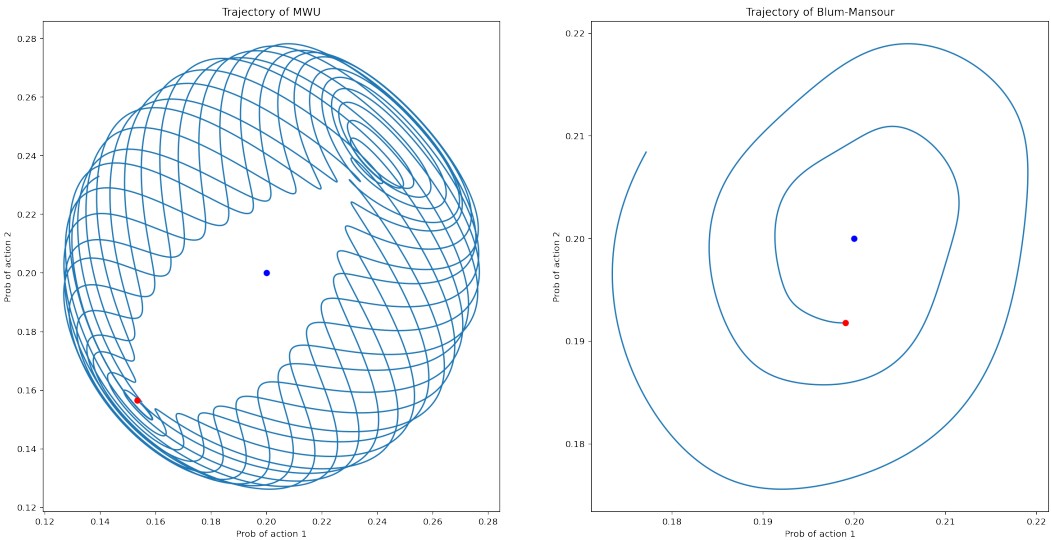

Figure 2: Trajectories of Multiplicative Weights and Blum-Mansour for Rock-Paper-Scissors-Lizard-Spock (a the 5-strategy generalization of Rock-Paper-Scissors).

- If Alice and Bob are running no-regret algorithms in a general game $G$, their strategies will have time-averaged convergence to a coarse correlated equilibrium of $G$.

- If Alice and Bob are running no-swap-regret algorithms in a general game $G$, their strategies will have time-averaged convergence to a correlated equilibrium of $G$.

- There exist zero-sum games $G$ (including Rock-Paper-Scissors of Example 1) where if Alice and Bob run certain no-regret algorithms (e.g. Multiplicative Weights) Alice and Bob's strategies will not have last-iterate convergence to a coarse correlated equilibrium.

## 3   Convergence of Symmetric Swap Regret in Symmetric Zero Sum Games

There exist zero-sum games $G$ (including Rock-Paper-Scissors of Example 1) where if Alice and Bob run certain no-regret algorithms (e.g. Multiplicative Weights) Alice and Bob's strategies will not have last-iterate convergence to a equilibrium. In the left side of Figure 1 plot the trajectory of Multiplicative Weights when Alice and Bob have the same initialization. The blue dot in the middle corresponds to the Nash equilibrium. While the strategies played average to the equilibrium, they never converge there.

In this section, we will show that for almost all *symmetric*, *zero-sum* games $G$, if Alice and Bob employ the same no-swap-regret learning algorithms with symmetric initializations they will have frequent-iterate convergence to a Nash equilibrium of $G$. In the right side of the same figure, we show the Blum-Mansour trajectory converging to equilibrium. In Appendix A, we show via a simple counter-example that symmetric initializations are necessary to achieve such strong convergence guarantees.

Our main tool is the following technical lemma, which shows that time-averaged convergence of symmetric action profiles to a product distribution implies frequent-iterate convergence to the same distribution.

**Lemma 1.** *Let $x_1, x_2, \ldots$ be a sequence of elements in $\Delta_N$ such that*

$$\lim_{T \to \infty} \left\| \left( \frac{1}{T} \sum_{t=1}^{T} x_t \otimes x_t \right) - (y \otimes y) \right\| = 0$$

*for some element $y$ of $\Delta_N$ (i.e., the profiles $x_t \otimes x_t$ time-averaged converge to $y \otimes y$). Then it is the case that for any $\varepsilon > 0$,*

$$\lim_{T \to \infty} \Pr_{t \leq T} \left[ \| (x_t \otimes x_t) - (y \otimes y) \| > \varepsilon \right] = 0.$$

*(i.e., the profiles $x_t \otimes x_t$ frequent-iterate converge to $y \otimes y$).*

*Proof.* For any $y \in \Delta_N$, we will show that there exists a linear functional $L_y : \mathbb{R}^N \otimes \mathbb{R}^N \to \mathbb{R}$ with the property that, among all elements of the form $x \otimes x$ with $x \in \Delta_N$, $L_y$ is uniquely minimized at $y \otimes y$. As a consequence, this implies that that there exists a $\delta > 0$ such that if

$$\| (x \otimes x) - (y \otimes y) \| > \varepsilon,$$

for some element $x \in \Delta_N$, then $L_y(x \otimes x) - L_y(y \otimes y) > \delta$. In particular, if for some $T$ we have that

$$\Pr_{t \leq T} \left[ \| (x_t \otimes x_t) - (y \otimes y) \| > \varepsilon \right] \geq \gamma,$$

then we consequently have that

$$\frac{1}{T} \sum_{t=1}^{T} \left( L_y(x_t \otimes x_t) - L_y(y \otimes y) \right) \geq \gamma \delta,$$

and in turn that

$$\left\| \left( \frac{1}{T} \sum_{t=1}^{T} (x_t \otimes x_t) \right) - (y \otimes y) \right\| \geq \gamma \delta \| L_y \|_*,$$

(where $\| L_y \|_*$ is the dual norm of the linear functional $L_y$ and is bounded below by some constant). This directly implies the lemma statement (it is impossible for the second quantity to approach zero as $T$ goes to infinity without the first quantity approaching zero).

We now describe the linear functional $L_y$. Let $y = (y_1, y_2, \ldots, y_N)$ (and for a general $x \in \Delta_N$, let $x = (x_1, x_2, \ldots, x_N)$). Note that for any linear functional $L_y : \mathbb{R}^N \otimes \mathbb{R}^N \to \mathbb{R}$, the value of

$L_y(x \otimes x)$ will be a homogeneous quadratic polynomial over the $x_i$; conversely, any such polynomial can be implemented as a linear functional over $\mathbb{R}^N \otimes \mathbb{R}^N$. Moreover, since $\sum x_i = 1$, we can convert any (non-homogeneous) quadratic polynomial to a homogeneous one (e.g. transforming $x_1^2 + x_1 + 1$ to $x_1^2 + x_1 \sum x_i + (\sum x_i)^2$). It therefore suffices to find a quadratic polynomial over the $x_i$ that is minimized when $x = y$.

We begin with the case where all $y_i > 0$. In this case, we claim that the polynomial

$$L_y(x \otimes x) = \sum_{i=1}^{N} \frac{1}{y_i} x_i^2$$

satisfies these constraints. To see this, note that by Cauchy-Schwartz,

$$\sum_{i=1}^{N} \frac{1}{y_i} x_i^2 = \left( \sum_{i=1}^{N} \frac{x_i^2}{y_i} \right) \left( \sum_{i=1}^{N} y_i \right) \geq \left( \sum_{i=1}^{N} x_i \right)^2 = 1,$$

with equality only holding when $x_i^2 / y_i = \lambda y_i$ for some fixed $\lambda$. Since $x$ and $y$ both belong to $\Delta_N$, this is only possible when $\lambda = 1$ and $x = y$.

What if some of the $y_i$ equal zero? Without loss of generality, assume $y_i > 0$ for $1 \leq i \leq n$ and $y_i = 0$ for $n + 1 \leq i \leq N$. Now, consider the polynomial

$$L_y(x \otimes x) = \left( \sum_{i=1}^{n} \frac{1}{y_i} x_i^2 \right) - 2 \left( \sum_{i=1}^{n} x_i \right).$$

We begin by minimizing this expression subject to $\sum_{i=1}^{n} x_i = s$, for some $0 \leq s \leq 1$. In this case, the second term in the above polynomial is identically $-2s$, so it suffices to minimize the first term. Again applying Cauchy-Schwartz, we find the minimum of the first term occurs only when $x_i = sy_i$, where it equals $s^2$. The overall minimum of $L_y$ (subject to this constraint) is therefore $s^2 - 2s$. This is in turn uniquely minimized when $s = 1$ (and $x_i = 0$ for all $i > n$), and therefore this $L_y$ is uniquely minimized at $x = y$. $\qquad \square$

Next we show a *'generic'* symmetric zero-sum game has a unique correlated equilibrium and hence coincides with the Nash equilibrium of this game. First, we define what we mean by generic. We can identify the set of zero-sum games with $\mathbb{R}^{N \times N}$, where each element corresponds to an $N \times N$ payoff matrix for Alice. We say that a property $\mathcal{P}$ holds for a generic zero-sum game, if the set of points in $\mathbb{R}^{N \times N}$ for which $\mathcal{P}$ doesn't holds form a measure zero subset of $\mathbb{R}^{N \times N}$.

By this definition, the set of symmetric zero-sum games forms a measure zero subset of the set of zero-sum games, so we need a refined definition to describe a generic symmetric zero-sum game. We can identify the set of symmetric zero-sum games with with $\mathbb{R}^{N(N-1)/2}$ where each element corresponds to a skew-symmetric $N \times N$ payoff matrix for Alice. We say that a property $\mathcal{P}$ holds for a generic symmetric zero-sum game, if the set of points in $\mathbb{R}^{N \times N}$ for which $\mathcal{P}$ doesn't holds form a measure zero subset of $\mathbb{R}^{N(N-1)/2}$.

The uniqueness of a correlated equilibrium for zero-sum games is known for generic zero-sum games by combining results by Forges [29] and Bohnenblust, Karlin and Shapley [12]. But since symmetric zero-sum games are a measure zero subset of zero-sum games, this result does not directly extend. Below, we extend it to symmetric zero-sum games.

Finally, to complete the picture, we further establish that this generic uniqueness of correlated equilibria strongly does not extend to the case of coarse correlated equilibria, which are the limit points of (external) regret-minimizing algorithms.

**Lemma 2.** *Almost all (i.e., all but a measure zero set of) two-player symmetric zero-sum games have a unique correlated/Nash equilibrium.*

*Proof.* By [29] a zero-sum game has a unique correlated equilibrium if and only if it has a unique Nash equilibrium, thus it suffices to prove the generic uniqueness of Nash equilibria. Let $A$ be the skew-symmetric ($A^\top = -A$) payoff matrix corresponding to the symmetric zero-sum game. A Nash

equilibrium is called quasi-strict (sometimes also referred to as regular or quasi-strong [34]) if for all agents deviations to strategies outside their support result in a strict decrease of their payoff. By Corollary 3.4 in [36] we have that in any two agent game if all equilibria are quasi-strict then the number of equilibria is finite. Thus, if a symmetric zero-sum game has multiple equilibria, this implies the existence of a non-quasi-strict equilibrium/optimal strategy. Let $x$ be that non-quasi-strict mixed strategy and let $S_x$ denote its support, i.e., the set of strategies played in $x$ with positive probability and let $i$ be the index of the strategy not in $S_x$ such that deviating to that strategy from the symmetric $(x, x)$ Nash equilibrium still results into payoff of zero (since the game value of any zero-sum symmetric game is zero). This implies that both antisymmetric sub-matrices of $A$ defined by its restrictions to the sets $S_x$ and $S_x \cup i$ respectively have a zero eigenvalue where the corresponding eigenvector is the analogously restricted subvector of $x$. Thus, both of their determinants are equal to zero. However, at least one of them has even dimension. It is well known that the determinant of an even dimension skew-symmetric matrix is a non-trivial polynomial (and in fact is the square of a polynomial in its coefficients, see e.g. [39]). The entries of this submatrix correspond to the vanishing set of a non-trivial polynomial and therefore have Lebesgue measure zero. Thus, the set of all symmetric zero-sum games with unique Nash/correlated equilibrium has zero measure.

$\square$

Combining Lemma 1 and Theorem 2, we arrive at the result mentioned at the beginning of this section.

**Theorem 3.** *In a generic symmetric zero-sum game $G$, if Alice and Bob run identical no-swap-regret algorithms with the same initialization to play $G$ repeatedly, their joint strategy profiles will have frequent-iterate convergence to a Nash equilibrium of $G$. Furthermore, this result is* tight, *i.e., it is not possible to prove (last-iterate) convergence to Nash equilibrium.*

*Proof.* Since Alice and Bob are both running the same no-swap-regret algorithm, their strategy profiles stay identical and hence their joint strategy profile is of the form $x_t \otimes x_t$. Since no-swap-regret algorithms have time average convergence to correlated equilibrium, then there is a correlated equilibrium $\sigma$ of $G$ such that $\|\frac{1}{T} \sum_{t=1}^{T} x_t \otimes x_t - \sigma\| \to 0$. By Lemma 2, $\sigma$ is a Nash equilibrium, so we can write $\sigma = y \otimes y$. Now, we can apply Lemma 1 to obtain frequent iterated convergence to $\sigma$.

It is not possible to prove anything stronger, i.e., convergence to Nash, based on (symmetric) no-swap-regret learning because we can take any such no-swap-regret dynamics and interject for a vanishing fraction of the history some arbitrary symmetric play where those payoff inputs are ignored by the learning dynamics (e.g. the Blum-Mansour algorithm does not see these fictitious entries). This rare interleaving of the trajectory with noise does not significantly affect the swap regret analysis which will remain sublinear if the original dynamic is no-swap-regret but at the same time it suffices to destroy any hope of true last iterate convergence. $\square$

### 3.1 Differences with respect to External Regret

It is useful to consider which parts of the proof break when we move from swap to external regret. Both Lemma 1 and Lemma 2 no longer hold.

Consider for example the executions of MWU and BM in Figure 1. In both cases we have that $\frac{1}{T}(\sum_t x_t) \to x_{\text{Nash}} := (\frac{1}{3}, \frac{1}{3}, \frac{1}{3})$. If we look at the empirical joint distribution $\bar{\sigma}_T = \frac{1}{T}(\sum_t x_t \otimes x_t)$ we obtain the following:

$$\bar{\sigma}_T^{\text{BM}} = \begin{bmatrix} 0.111 & 0.111 & 0.111 \\ 0.111 & 0.111 & 0.111 \\ 0.111 & 0.111 & 0.111 \end{bmatrix} \qquad \bar{\sigma}_T^{\text{MWU}} = \begin{bmatrix} 0.120 & 0.105 & 0.105 \\ 0.105 & 0.120 & 0.105 \\ 0.105 & 0.105 & 0.120 \end{bmatrix}$$

Only in the swap regret algorithm we have $\bar{\sigma} \to x_{\text{Nash}} \otimes x_{\text{Nash}}$. Lemma 1 crucially relies on $\frac{1}{T}(\sum_t x_t \otimes x_t)$ converging a product distribution.

Lemma 2 also breaks when we replace CE (achieved by swap regret minimization) with CCE (achieved by external regret minimization):

**Lemma 4.** *Given any symmetric zero-sum game as long as its set of optimal strategies does not consist of a single pure (i.e., deterministic) strategy then it has a continuum of coarse correlated equilibria.*

*Proof.* In any such game there exists an optimal (Nash) strategy that is randomizing amongst at least two strategies. We will define the correlated distribution that applies positive probability only symmetric outcomes of the game (e.g, such as (Rock, Rock) (Paper, Paper) and (Scissors, Scissors) in the RPS game) such as its resulting marginal distribution corresponds to the Nash equilibrium strategy. The expected payoff of each agent in this distribution is equal to zero. Furthermore, any deviating strategy cannot result in positive payoff since the marginal distribution encodes a Nash equilibrium. Thus, the original distribution is a CCE. By taking convex combinations of these CCE and the Nash equilibrium we have that each such game has a continuum of CCE. □

An immediate corollary of Lemma 4 is that the rich, non-equilibrating behavior of MWU and other no-regret dynamics, even under symmetric initializations, shown for Rock-Paper-Scissors and Rock-Paper-Scissors-Lizard-Spock shown in Figures 1,2 should be common for many other symmetric games and regret-minimizing dynamics as well.

### 3.2 Asymmetric Zero Sum Games

We remark that the results don't generalize to asymmetric zero sum games. One example of such games is matching pennies where no-external regret dynamics like MWU are known not to exhibit last-iterate convergence and in fact diverge chaotically towards the boundary for all interior initializations, including all symmetric (non-Nash) initial conditions [8, 18]. For games with 2 actions per player, any no-external-regret algorithm is also no-swap-regret as we show in the following lemma.

**Lemma 5.** *For a game with* 2 *actions per player,* $\mathsf{SwapReg} \leq 2\,\mathsf{Reg}$.

*Proof.* Let $\{0, 1\}$ be Alice's actions in let $a_t, b_t \in \Delta_2$ be a sequence of strategies played by Alice and Bob. There are 3 non-trivial swap strategies $\pi : [2] \to [2]$ for Alice: $s \mapsto 0$, $s \mapsto 1$, $s \mapsto 1 - s$. The regret of using the first two swap strategies is bounded by Reg since they map to a constant action. Let $a_t[s]$ be the $s$-th component of $a_t \in \Delta_2$. Then, the regret of the third strategy $\pi$ is bounded by:

$$\sum_t G_A(a_t, b_t) - G_A(\pi(a_t), b_t) = \sum_t a_t[0](G_A(0, b_t) - G_A(1, b_t)) + \sum_t a_t[1](G_A(1, b_t) - G_A(0, b_t))$$

$$= \left[\sum_t G_A(a_t, b_t) - G_A(1, b_t)\right] + \left[\sum_t G_A(a_t, b_t) - G_A(0, b_t)\right] \leq 2\,\mathsf{Reg}$$

□

## 4 No-Swap-Regret Algorithms are Time-Asymmetric

One interesting feature of no(-external)-regret algorithms in two-player games is that the action they select at time $t$ can depend entirely on the average historical strategy played by the other player up to time $t - 1$, and not on any other information about how this strategy (or the player's own strategy) evolved over time.

In this section we show that it is impossible for a no-swap-regret learning algorithm to have this property. In fact, we prove the following more general statement.

**Theorem 6.** *Consider any learning algorithm* $\mathcal{A}$ *which decides what action to take on behalf of Alice in a game* $G$ *at round* $t$ *via a* symmetric *function* $A_t(b_1, b_2, \ldots, b_{t-1})$ *of Bob's mixed strategies up until* $t - 1$ *(here symmetric means that the function is unchanged for any permutation of the inputs). Then there exists a game* $G$ *and a sequence of play for Bob where Alice incurs* $\Omega(T)$ *swap regret.*

As mentioned, many no-external-regret algorithms (such as multiplicative weights, and follow-the-regularized leader) have the property that each $A_t$ can be written as a function of the average $\frac{1}{t-1}\sum_{s=1}^{t-1} b_s$, and hence are symmetric.

We provide a sketch of the proof of Theorem 6, deferring all details and proofs of lemmas to Appendix B. We will consider the game $G$ given by the three-action generalization of Matching Pennies. Specifically, Alice and Bob will both have 3 actions, and $G_A(i, j)$ equals 1 if $i = j$ and 0 otherwise (since we will specify Bob's sequence of actions adversarially, his payoff $G_B$ is irrelevant).

We will construct an adversarial sequence of actions for Bob where in each round Bob plays one of the three pure actions in $\Delta_3$ (i.e., $b_t \in \{e_1, e_2, e_3\}$). For each $i \in [3]$, let $n_{t,i}$ equal the number

of rounds $s \leq t - 1$ where Bob played action $i$. Since $A_t$ is a symmetric function of the $b_s$ for $1 \leq s \leq t - 1$, we can write $A_t$ as a function $A_t(n_{t,1}, n_{t,2}, n_{t,3})$. Moreover, since we must have $n_{t,1} + n_{t,2} + n_{t,3} = t - 1$, we can summarize the entire learning algorithm with a single 3-variable function $A(n_1, n_2, n_3) : \mathbb{Z}_{\geq 0}^3 \to \Delta_3$ satisfying

$$A(n_1, n_2, n_3) = A_{n_1+n_2+n_3+1}(n_1, n_2, n_3).$$

Our main strategy will be to show that if $A$ has no swap regret (or even no external regret), on average $A$ must be very close to the "follow the leader" strategy, which puts all the weight on action $i$ if $n_i$ is significantly larger than the other $n_j$. We formalize this in the following lemma (which employs a result of [25] on the number of mistakes necessary in certain card guessing games):

**Lemma 7.** *Assume that the algorithm $\mathcal{A}$ guarantees that Alice incurs at most sublinear external regret, i.e., $\mathsf{Reg}_A = o(T)$. Then for any $L \geq T/100$ and $n_1, n_2 \leq n_3 \leq T - L$, we have that*

$$\frac{1}{L} \sum_{m=n_3+1}^{n_3+L} A(n_1, n_2, m)_3 = 1 - o(1).$$

Lemma 7 shows that $\mathcal{A}$ mostly plays the leader (i.e., highest-utility) action in any sufficiently long segment of rounds where Bob is playing the leader action. We would also like to show that $\mathcal{A}$ mostly plays the leader action in stretches where Bob is playing some other fixed action. The following lemma gives a weak form of this claim (but that will be sufficient for proving Theorem 6).

**Lemma 8.** *Fix any $L, L' \geq T/100$ and let $n_2 \geq \min(n_1, L')$. Then there exists an $n_3 \in [n_2, n_2 + L]$ such that*

$$\frac{1}{L'} \sum_{m'=n_2-L'}^{n_2-1} A(n_1, m', n_3)_3 = 1 - o(1).$$

With Lemmas 7 and 8 (and their symmetric counterparts), we can construct a sequence of play for Bob where Alice incurs high swap regret. Roughly, this sequence of play proceeds as follows.

- First Bob plays action 1 for approximately $T/3$ rounds. By Lemma 7, we can guarantee that Alice plays action 1 for most of these rounds.

- Bob then plays action 2 for approximately $T/3$ rounds. By the guarantee of Lemma 8, Alice will still play action 1 for most of these rounds.

- Bob then plays action 3 for the remaining rounds. Again by applying Lemma 8, we can guarantee that Alice will play action 2 for most of these rounds.

It is straightforward to check that Alice incurs linear swap regret in the above trajectory – Alice would improve her expected utility by $\Omega(T)$ if she played action 3 every time she played action 2. The details of this proof are deferred to Appendix B.

## 5 Conclusion

In this paper, we study the role of symmetry in the behavior of no-swap regret dynamics. In our first result, we show that no-swap-regret dynamics in self-play in symmetric zero-sum games lead to converge in a strong "frequent-iterate" sense to the Nash equilibrium. Specifically, in all but a vanishing fraction of the rounds, the players must play a strategy profile close to a symmetric Nash equilibrium. Furthermore, we show that the power of no-swap-regret dynamics comes at a cost of imposing a time-asymmetry on its inputs. Specifically, any such algorithm, unlike no-external regret dynamics, must apply a time-asymmetric function over the set of previously observed rewards.

The interplay between symmetry, (external/swap) regret and learning in games emerges as as interesting direction for future work. One particularly interesting such direction would be to explore generalizations of our results beyond two player games.

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

# A No-swap regret dynamics with asymmetric initial conditions do not converge to Nash equilibrium

We show via a simple counter-example that symmetric initializations are necessary for convergence to Nash. Specifically, in Figure 3 we show the trajectory of two players running the Blum-Mansour no-swap-regret algorithm against each other, initialized with asymmetric starting conditions in the symmetric zero-sum game of Rock-Paper-Scissors. Their behavior does not converge to Nash, whereas as we have seen in Figure 1, symmetric initialization in the same game would have led to convergence.

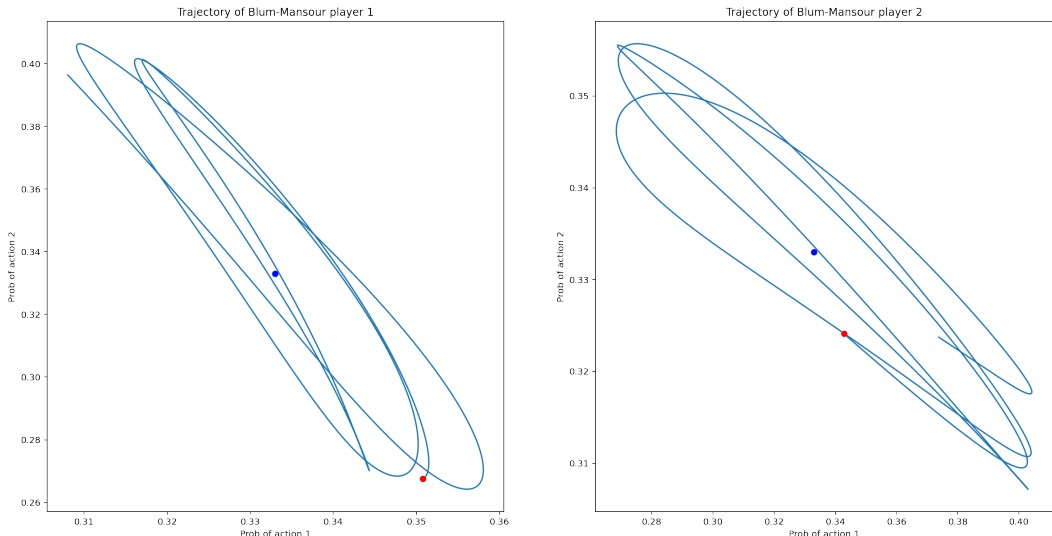

Figure 3: This figure shows the trajectory of two players running the Blum-Mansour no-swap-regret algorithm against each other initialized with asymmetric starting conditions. Unlike the symmetric dynamics, these do not ultimately converge to the unique symmetric Nash equilibrium (the blue point).

# B Omitted proofs of Section 4

## B.1 Proof of Lemma 7

*Proof of Lemma 7.* Define $W = \sum_{m=n_3+1}^{n_3+L} A(n_1, n_2, m)_3$. We will present a distribution $\mathcal{D}$ over action sequences of length $T$ for Bob such that the expected regret $\text{Reg}_A$ that Alice incurs when playing against a randomly drawn sequence from $\mathcal{D}$ is at least $(L - W) - O(\sqrt{T})$. If the algorithm $\mathcal{A}$ has the guarantee that $\text{Reg}_A = o(T)$, then this implies that we must have $W = L - o(T)$, from which the theorem statement follows.

We will construct the distribution $\mathcal{D}$ as follows:

- For the first $n_1 + n_2 + n_3$ rounds, Bob will play a uniform random sequence that includes action 1 $n_1$ times, action 2 $n_2$ times, and action 3 $n_3$ times.

- In the next $L$ rounds, Bob will always play action 3.

- Finally, in the last $T' = T - (n_1 + n_2 + n_3)$ rounds, Bob will play a uniform random sequence that includes action 1 $T'/3$ times, action 2 $T'/3$ times, and action 3 $T'/3$ times.

Note that for any such sequence, the best action in hindsight for Alice is action 3, which achieves utility exactly $n_3 + L + T'/3$. To compute Alice's expected regret, it suffices to compare Alice's expected utility to this quantity.

To bound Alice's optimal expected utility, we will make use of the following result of Diaconis and Graham [25]. Consider a game where there is a uniformly shuffled deck of $N = n_1 + n_2 + n_3$ cards with $n_1$ cards labeled 1, $n_2$ cards labeled 2, and $n_3$ cards labeled 3. In this game, the player must repeatedly guess the label of the card on the top of the deck, at which point it is revealed to the player and discarded. Then Diaconis and Graham show that the expected number of correct guesses of the player is at most $\max(n_1, n_2, n_3) + O(\sqrt{N})$ (in fact in Theorem 3 of [25] provides bounds that apply to any number of labels, but we only need this specific consequence).

Note that in the first $n_1 + n_2 + n_3$ rounds, Alice is facing essentially this exact game (since she gains utility 1 when she matches the action of Bob, and 0 otherwise). Therefore Alice's expected utility in the first segment of the rounds is at most $n_3 + O(\sqrt{n_1 + n_2 + n_3}) = n_3 + O(\sqrt{T})$. Similarly, in the last segment of rounds, Alice's expected utility is at most $T'/3 + O(\sqrt{T})$. Finally, in the middle segment of rounds, Alice's utility is exactly $W$ (as this is the total weight she places on action 3). It follows that Alice's expected regret is at least $(L - W) - O(\sqrt{T})$, as desired. $\qquad \square$

## B.2 Proof of Lemma 8

*Proof of Lemma 8.* By applying Lemma 7 $L$ times, we have that:

$$\frac{1}{L' \cdot L} \sum_{m'=n_2-L'}^{n_2-1} \sum_{m=n_2}^{n_2+L} A(n_1, m', m)_3 = 1 - o(1).$$

By switching the order of summation, this implies that there must exist a fixed value of $m \in [n_2, n_2 + L]$ such that

$$\frac{1}{L'} \sum_{m'=n_2-L'}^{n_2-1} A(n_1, m', m)_3 = 1 - o(1).$$

We can take $n_3$ to be this value of $m$. $\qquad \square$

With Lemmas 7 and 8 (and their symmetric counterparts), we can construct a sequence of play for Bob where Alice incurs high swap regret.

## B.3 Proof of Theorem 6

*Proof of Theorem 6.* Fix $L = T/100$. Bob will begin by selecting an $n_1 \in [T/3, T/3 + T/1000]$ such that:

$$\frac{1}{T/3} \sum_{m'=1}^{T/3} A(n_1, m', 0)_3 = 1 - o(1).$$

(Such an $n_1$ is guaranteed to exist by a symmetric variant of Lemma 8). Bob will then play action 1 for $n_1$ rounds followed by action 2 for $T/3$ rounds.

Bob will then select a value $n_2 \in [T/3 + T/1000, T/3 + 2T/1000]$ such that:

$$\frac{1}{(T/3) - (T/100)} \sum_{m'=1}^{T/3-T/100} A(n_1, n_2, m')_3 = 1 - o(1).$$

(Again, such an $n_2$ is guaranteed to exist by a symmetric variant of Lemma 8). Bob will then play action 2 for $n_2 - T/3$ rounds, and action 3 for the remaining rounds.

What does Alice do against this sequence of play of Bob? We break this down segment by segment:

- First Bob plays action 1 for $n_1$ rounds, moving the state from $(0, 0, 0)$ to $(n_1, 0, 0)$. Since $n_1 > T/1000$, Lemma 7 implies that Alice plays action 1 for $1 - o(1)$ of these rounds.

- Bob then plays action 2 for $T/3$ rounds, moving the state to $(n_1, T/3, 0)$. By the guarantee of Lemma 8, Alice will still play action 1 for $1 - o(1)$ of these rounds.

- Bob then plays action 2 for $n_2 - T/3$ more rounds, moving the state to $(n_1, n_2, 0)$. This is at most $T/500$ rounds, which will be negligible in our final swap regret computation.

- Bob then plays action 3 for $(T/3) - (T/100)$ rounds, moving the state to $(n_1, n_2, T/3 - T/100)$. By the guarantee of Lemma 8, Alice will play action 2 for $1 - o(1)$ of these rounds.

- Bob then plays action 3 for the remaining rounds. This is at most $T/100$ rounds, which will be negligible in our final swap regret computation.

The key observation is that Alice would significantly improve her expected utility (by $\Omega(T)$) by playing action 3 every time she played action 2, and therefore Alice has linear swap regret. To see this, note that Alice plays action 2 for at least $(1 - o(1))((T/3) - (T/100)) \geq T/4$ rounds when Bob is playing action 3. On the other hand the number of rounds where both Alice and Bob play action 2 is at most the $T/500$ rounds in the third segment. Therefore the expected gain from switching from action 2 to action 3 is at least $T/4 - T/500 = \Omega(T)$, and therefore the algorithm $\mathcal{A}$ incurs $\Omega(T)$ swap regret. $\qquad\square$

