# OpenReview forum: "Convergence of No-Swap-Regret Dynamics in Self-Play"
_NeurIPS.cc/2024/Conference — NeurIPS 2024 poster_

### Official Review · Reviewer_NEnG · 2024-06-24

**Soundness:** 3
**Presentation:** 3
**Contribution:** 2
**Rating:** 4
**Confidence:** 3

**Summary:**

The paper proves that in "almost all" symmetric zero-sum two-player games (excluding a measure zero set), if both players apply the same no-swap-regret algorithm with the same initialization, then the players' iterates satisfy frequent-iterate convergence to Nash-equilibrium.

The main two lemmas forming the proof are: Lemma 1, which shows that time-average convergence of symmetric action profiles leads to frequent-iterate convergence; and Lemma 2, which demonstrates that almost all symmetric zero-sum games have a unique correlated equilibrium (CE), thereby implying a unique Nash equilibrium (NE).

Finally, the paper shows that no-swap-regret algorithm cannot be determined entirely by cummulative cost induced by the opposing's player strategy (this is unlike to some no-regret algorithm such as vanilla FTRL). This result is outside the scope of games (the other player is adverserial and induce some arbitrary loss).

**Strengths:**

* The paper is quite self-contained and gives a good background on game theory and the different notions of convergence to equilibria.
* Overall, the paper is well written, and most of the proofs are quite clear and easy to follow.

**Weaknesses:**

The main limitation of this paper is that while the it demonstrates frequent-iterate convergence to Nash equilibrium for the family of no-swap-regret algorithms under specific conditions, there are existing algorithms that achieve last-iterate convergence to Nash equilibrium, in any zero-sum game, without the need for the symmetry and uniqueness constraints assumed here. In this context, the significance of the paper appears somewhat limited, as it establishes a much weaker guarantee than what is generally achievable.

It should also be noted that the abstract is somewhat misleading: The autors say that *no-swap-regret dynamics are guaranteed to converge in a strong, last-iterate sense to the Nash equilibrium*. This is not true, the dynamics will give frequent-iterate convergence. In fact the authos explicitly state in Thorem 3 that *"it is not possible to prove (last-iterate) convergence to Nash equilibrium"*.

************************************

#### Typos and other comments

Line 101: There are also exist algorithms with regret $\tilde O (1)$.

Line 155: missing citations

Line 247: unique -> multiple?

Line 249: Theorem -> Lemma

Line 251: identical identical -> identical

Line 300: It is a feature of only **some** no-regret algorithms

**Questions:**

* Do you think it is possible to show a non-asymptotic guarantee in this setting? I.e., to show the convergence rate in terms of the algorithm's regret guarantee?
* In line 246: what do you mean by the "*vanishing set* of a non-trivial polynomial"?

**Limitations:**

See Weaknesses

---

> ### Author Rebuttal · Authors · 2024-08-07
>
> Thank you for your review! We are happy to hear that you found our paper to be self-contained, well written and easy to read!
>
> We view the main contribution of our paper mapping the landscape of which algorithms/settings it is possible to obtain last-iterate convergence. Note that in terms of algorithms our result is very general as it captures widely used algorithms such as Blum-Mansur and Stolz-Lugosi as well as newly developed ones [18,42].
>
> We believe the class of symmetric zero-sum games is central to many Machine learning  applications, e.g., [1-4]. The uniqueness assumption is somewhat technical and easily achieved by any tiny perturbation of payoffs. A technique commonly used in practice is  “self-play” — i.e. we only keep a single agent that competes against themselves. We observe that our analysis exactly corresponds to self-play using no swap regret algorithms. In many situations we may want to use no swap regret algorithms for other reasons such as robustness to manipulation or a correlated-equilibrium-like structure. Our result shows that in that case, we obtain last-iterate convergence for free.
>
> We opted for not mentioning novel technical terminology of frequent iterate convergence in the abstract to improve readability, but we are happy to change that if the reviewer thinks it will improve the paper. We made sure to clarify what we mean in the intro and explain in simple language why this result is the strongest possible without further assumptions.
>
>
> Q1. See our comment in response to Q1 of Reviewer r1mG re: rates of convergence.
>
> Q2. A non-trivial polynomial is a polynomial that does not always evaluate to zero. The vanishing set of a (multi-variate) polynomial is the set of input values such that its value becomes equal to zero.
>
> We hope that our answers addressed your initial questions and if so that you would consider increasing your score. Thank you again for your time and interesting questions!

---

> > ### Comment · Reviewer_NEnG · 2024-08-08
> >
> > Thank you for your response and the clarifications provided. I have read all the reviews and your rebuttals. At this point, I will retain my initial score.
> >
> > Regardless of the final decision on the acceptance of the paper to NeurIPS, and although this is not my primary concern, I encourage the authors to include the rate of convergence in terms of the swap regret guarantee in future versions of the paper. Additionally, I recommend being more precise in the abstract regarding the claim on last-iterate convergence - as the body of the paper demonstrates, this claim is not true.

---

### Official Review · Reviewer_r1mG · 2024-07-01

**Soundness:** 3
**Presentation:** 3
**Contribution:** 2
**Rating:** 6
**Confidence:** 4

**Summary:**

This paper studies the convergence properties of no-swap-regret learning dynamics in symmetric two-player zero-sum games. The paper's main result is that in almost all symmetric zero-sum games with symmetric initializations, if both agents run identical no-swap-regret algorithms, their joint strategy profiles will have frequent-iterate convergence to a Nash equilibrium. This result is enabled by showing (1) almost all symmetric zero-sum games have a unique Nash equilibrium and thus a unique correlated equilibrium; (2) time-averaged convergence of symmetric strategy profiles to *a* product distribution implies frequent-iterate convergence to the same distribution. Here, frequent-iterate convergence to some strategy profile $\sigma$ means that almost all joint strategy profiles will be arbitrarily close to $\sigma$ as time goes on. Moreover, this convergence result does not generalize to asymmetric games. This paper also shows that any no-swap-regret algorithm must play time-asymmetric functions over the set of history rewards. In contrast, specific no-external-regret algorithms, like Multiplicative Weights Update, are based on symmetric functions.

**Strengths:**

This paper studies an interesting question of the convergence properties of no-swap-regret learning in two-player zero-sum games. Although we know no-swap-regret dynamics diverge for general zero-sum games, this paper shows a frequent-iterate convergence to a Nash equilibrium in symmetric zero-sum games with symmetric initialization. These assumptions on the game and initializations are very strong, but the paper also shows they are necessary for convergence. The proofs in the paper are also simple.

**Weaknesses:**

1. In the introduction, the paper motivates the study by "*Although it is possible to stabilize learning dynamics in zero-sum games using, e.g., optimistic variants of MWU, such results leave something to be desired as they presuppose that the agents coordinate to use a specific instantiation of a learning algorithm.* I thought the result would allow the two agents to use any algorithms so long as they are no-swap-regret. However, the convergence result still requires both agents to coordinate and use an identical algorithm. Moreover, this result also requires both agents to have symmetric initializations, while no-external-regret algorithms like OGDA and OMWU do not. Why is this an advantage over using OGDA/OMWU?
2. There are no results on the rates of convergence.
3. Typos:
   1. Line 104: "$G_A(a_t, b_t) = G_B(b_t, a_t)$" should be $G_A(a_t, b_t) = G_A(b_t, a_t)$?
   2. Line 155: Missing reference
   3. Line 173: change to "an equilibrium"
   4. Line 216: "with with"
   5. Line 251: "identical identical"

**Questions:**

1. Is it possible to prove a convergence rate to a Nash equilibrium in terms of the swap regret of the algorithm?
2. In the abstract, it is claimed that relaxing the symmetric initializations would destroy the convergence result. But I do not find this result in the main body. Does the convergence hold regardless of the initialization as OGDA/OMWU?
3. What are the convergence properties when both agents use non-identical swap-regret algorithms in symmetric zero-sum games? Is there a counterexample that they do not converge in this case?

**Limitations:**

See weakness.

---

> ### Author Rebuttal · Authors · 2024-08-07
>
> We thank you for your support. We are very happy to hear that you found the subject matter interesting and that you appreciate the simplicity of our proofs!
>
>
> The reviewer asks “why is this an advantage over using OGDA/OMWU?” We don’t think OGDA/OMWU is inherently better or worse than NoSwapRegret algorithms, it really depends on the application. Our goal is to expand the set of algorithms/settings where it is known that we can obtain last-iterate convergence. If last-iterate convergence is the only goal, then OGDA/OMWU may be sufficient. But if we have other goals like robustness to manipulation or a correlated-equilibrium-like structure, then we may prefer to use no swap regret instead. Our result shows that in that case, we obtain (essentially) last-iterate convergence for free.
>
> While the symmetric initialization may appear restrictive at first, it is commonly used in practice in what we call “self-play” — i.e., we only keep a single agent that competes against themselves (see [1-3] for a handful examples in different applications). Hence, understanding self-play behavior can lead to new insights in many such real world applications. An important emerging application is “AI Debate” (see e.g. [4] and references therein) a zero-sum game with a very large strategy space in the area of AI Safety for which novel no-swap regret algorithms have been recently developed. For symmetric versions of those games, no swap regret provides important benefits since Nash equilibria can be computed without averaging over a large number of strategies which are not easy to keep in memory.
>
> Q1. Although we didn’t explicitly consider convergence rates in our paper, we immediately inherit the known time-averaged convergence rates to the space of CE (since the unique CE of the game is this symmetric NE that we are converging to). That is, if we have a learning algorithm with swap regret bounded by R(T), we should approach this NE at a rate of ~R(T)/T. We are happy to add a discussion of rates to the paper. It is an interesting question to understand if variations on our analysis can lead to rates stronger than those naively obtained through the existing time-average convergence bounds.
>
> Q2.  Asymmetric initializations destroy the convergence theorem:  In terms of establishing that relaxing the symmetric initializations destroys the convergence result it suffices to run a single simulation of e.g. Blum-Mansour algorithm in Rock-Paper-Scissors with asymmetric initial state. Since this is relatively easy to verify we had initially decided against adding such a figure, however, if you think this will strengthen our paper, we will be happy to add it. (See figure attached to the general response).
>
>  Q3. Non-identical swap regret algorithms in symmetric zero-sum games destroy the convergence theorem: Again a single simulation would suffice to establish this result. E.g. Running Blum-Mansour algorithm with MWU and different rates of decreasing step-sizes would suffice. Interestingly, one can argue that this question reduces to Q2 hence the above counter-example suffices. We can interpret Blum-Mansour with MWU and different initializations as running two different variants of Blum-Mansour each of them running with a slightly different FTRL algorithm where the regularizer has absorbed their distinct initializations.
>
>
> We hope that our answers addressed your questions above and if so you would consider increasing your score accordingly. Thank you again for your interesting questions!
>
> [1] Lanctot, Marc, et al. Section A.3 in "A unified game-theoretic approach to multiagent reinforcement learning." Advances in neural information processing systems 30 (2017).
>
> [2] McAleer, S., Lanier, J. B., Wang, K., Baldi, P., Fox, R., & Sandholm, T. (2022). Self-play psro: Toward optimal populations in two-player zero-sum games. arXiv preprint arXiv:2207.06541.
>
> [3] Swamy, Gokul, Christoph Dann, Rahul Kidambi, Zhiwei Steven Wu, and Alekh Agarwal. "A minimaximalist approach to reinforcement learning from human feedback." arXiv preprint arXiv:2401.04056 (2024).
>
> [4] Chen, Xinyi, Angelica Chen, Dean Foster, and Elad Hazan. "Playing large games with oracles and ai debate."

---

> > ### Comment · Reviewer_r1mG · 2024-08-09
> > **Response by the Reviewer**
> >
> > I thank the authors for answering my questions. After reading all the reviews and rebuttals, I decided to increase my score by 1.
> >
> > Additional Comment: I believe It is better to define and use the "frequent iterate convergence" terminology in the abstract since it could be misleading as it differs greatly from the literature's traditional notion of "last-iterate convergence." I suggest adding the sentence in the main body (Line 148) to the abstract: "Frequent-iterate convergence means that, as time goes on, almost all joint strategies profiles the players play will be arbitrarily close to a Nash equilibrium." Also, I found the notion of "frequent iterate convergence" to be similar to the "time-average convergence" defined in [DHLZ22]: the fraction of rounds where bidders play a Nash equilibrium approaches 1 in the limit (see their abstract). Their "time-average convergence" definition is also non-standard and is not the same as "average-iterate convergence" in the literature. Thus, I think adding at least one informal definition to the abstract would help the reader understand the result better.
> >
> > Deng, X., Hu, X., Lin, T., & Zheng, W. (2022, April). Nash convergence of mean-based learning algorithms in first price auctions. In Proceedings of the ACM Web Conference. 2022

---

### Official Review · Reviewer_xwUS · 2024-07-10

**Soundness:** 3
**Presentation:** 3
**Contribution:** 3
**Rating:** 7
**Confidence:** 4

**Summary:**

This submission studies no-swap-regret dynamics in two-player zero-sum games. In particular, they make the novel observation that no-swap-regret dynamics provably converge in a last-iterate-like sense ("frequent iterate convergence") to Nash equilibria in symmetric two-player zero-sum games. In order to show this result, the authors prove that time-averaged convergence of symmetric action profiles implies frequent-iterate convergence to the same distribution. This, combined with the fact that "almost all" (in a measure-theoretic sense) two-player symmetric zero-sum games have a unique Nash/correlated equilibrium, yields the authors' main result.

The authors also show that this result does not hold for no-external-regret dynamics, and that it does not hold for no-swap-regret dynamics outside the class of symmetric two-player zero-sum games in general.

**Strengths:**

The fact that no-swap-regret dynamics converge to Nash equilibria in a near-last-iterate sense is interesting, and is the first result of its kind to the best of my knowledge. (As the authors point out, one can get last-iterate convergence convergence to equilibria if all players use "optimistic" no-regret learning algorithms, but this presupposes that all players coordinate ahead of time to use a specific instantiation of a learning algorithm.)

This result is important because it gives a reason why one may prefer to play no-swap-regret dynamics over no-external-regret dynamics in (a subset of) two-player zero-sum games. Moreover, the results are interesting for using learning dynamics for the purposes of equilibrium computation: computing the time-average distribution over actions for each player is no longer required in symmetric two-player zero-sum games if both players are instantiated with no-swap-regret dynamics.

The authors' other results are complementary, in the sense that they answer several obvious questions that the reader may have after reading the authors' main result. (E.g., "Do these results extend to no-external-regret dynamics?", "Do these results extend beyond symmetric games?")

Finally, the theoretical analysis is presented in a straightforward manner and was relatively easy for me to follow.

**Weaknesses:**

While the authors main result is interesting, I would have like to have seen it fleshed out a bit more, especially with respect to frequent-iterate convergence vs last-iterate convergence. (See 'Questions' for more details.)

The empirical results in Figures 1 and 2 are cool, and it would have been nice to see a more robust set of experiments/simulations.

**Questions:**

If you fix the swap regret dynamics of each player (e.g., the Blum-Mansour algorithm instantiated with MWU), can you say anything about the last-iterate convergence of the dynamics (beyond the frequent-iterate convergence shown in the submission)? Can you give a necessary/sufficient condition for last-iterate convergence of no-swap-regret dynamics?

Do your results carry over to symmetric extensive-form games with incomplete information?

Comment: The writing is a bit sloppy in places, and the submission could benefit from another pass by the authors. For example, there are typos/missing references in footnote 1, line 155, and line 166.

**Limitations:**

The authors have adequately addressed the limitations of their work.

---

> ### Author Rebuttal · Authors · 2024-08-07
>
> We thank you for your support! We are very encouraged to hear that you find our results to be both important as well as interesting!
>
> Currently, we do not have any stronger analysis for the case of specific no-swap regret algorithms such as e.g., the Blum-Mansour initiated with MWU. Empirically, we observe last-iterate convergence and so we believe that this holds. Overall, we agree that each well studied instance of no-swap algorithms deserves individual attention and we are happy to explicitly pose this an open question for future work.
>
> Our results should extend to any game that has even an inefficient interpretation as a 2-agent normal form game (e.g. extensive form games or Bayesian games with private information). However note that we do assume we operate in a full-information setting (where the full counterfactual of losses is available to us) -- it is not obvious how to extend these results to e.g. the case of bandit feedback (it is not even clear what “symmetric dynamics” should mean in such partial-information settings).

---

> > ### Comment · Reviewer_xwUS · 2024-08-08
> >
> > Thanks for your reply. I will maintain my score.
> >
> > Moreover, I disagree with Reviewer NEnG that the significance of this paper is limited because there are other algorithms which are known to converge in a last-iterate sense to NE in two-player zero-sum games. I agree with the authors that there is value in discovering new properties of existing/popular algorithms for playing in games.

---

### Official Review · Reviewer_b4Uc · 2024-07-11

**Soundness:** 3
**Presentation:** 3
**Contribution:** 3
**Rating:** 6
**Confidence:** 4

**Summary:**

The paper studies the convergence of no-swap-regret dynamics in zero-sum games. In particular, it is shown that in almost all symmetric zero-sum games and under a symmetric initialization, no-swap-regret dynamics are guaranteed to converge in a last-iterate sense to a Nash equilibrium; all of the previous assumptions are necessary. This establishes a stark separation between external and swap regret as a predictor for the day-to-day behavior of the dynamics. Furthermore, they complement the previous result by showing that any no-swap-regret algorithm must rely on an asymmetric function of the previously observed rewards.

**Strengths:**

One important question in the theory of learning in games is to characterize learning algorithms which guarantee iterate-convergence to Nash equilibria in zero-sum games. Many common algorithms, such as MWU, do not have that property. While there are some positive results in the literature mostly concerning optimistic mirror descent, they are rather isolated and do not provide a more appealing characterization. This paper provides an interesting such class of algorithms: those minimizing swap regret. I believe that this is an important contribution, and further complements a recent line of work that illuminates other strong properties of swap regret. The result is also surprising; at first glance, studying no-swap-regret dynamics in zero-sum games seems counterintuitive because of the connection between CCE and NE in such games. The necessity of using a time-asymmetric function for minimizing swap regret is also interesting, and appears to be new. Overall I believe that this paper would be well-received from the learning in games community at NeurIPS.

**Weaknesses:**

One weakness is that the proof of the main result is fairly straightforward and somewhat uninformative, in that it relies on an elementary observation (Lemma 1), perhaps suggesting that the result is not particularly deep.

In terms of the organization of the paper, I feel that not enough gravity has been given to Section 4. The proof in Appendix A appears to contain a number of interesting ideas which are not discussed in the main body. One suggestion would be to expand that section, perhaps at the cost of deferring to the appendix some of the earlier proofs that are not particularly interesting, such as those of Lemmas 1 and 2. It would also strengthen the paper if the authors could explain further the significance of the main result in Section 4, potentially discuss implications and applications. It also seems that the results in Section 3 are not particularly well-connected with the result of Section 4; perhaps the authors can address this point in the revision as well.

**Questions:**

See above.

**Limitations:**

The authors have adequately addressed the limitations.

---

> ### Author Rebuttal · Authors · 2024-08-07
>
> We thank you for your support! We particularly appreciate that you point out that you find our contribution to be both important as well as surprising!
>
> We believe that the relative simplicity of our proof (when e.g. compared against the analysis of optimistic mirror descent) should be seen as an important benefit of our analysis. Not only is this result surprising and important but it is also easy to articulate and parse, while at the same time being applicable to a wide range of no-swap regret algorithms.
>
> Furthermore, we very much appreciate your acknowledgement that the proofs of Section 4 that have been deferred to the Appendix have a number of interesting ideas! We will follow your suggestions to move these points in the main body as well as overall implement your helpful recommendations to further enhance our paper.   One interesting implication of Section 4 is that although the vector of aggregate payoffs/probabilities of historical play is a natural and concise state space for no-regret algorithms such as FTRL, it does not suffice for swap-regret, raising interesting questions about exploring the Pareto frontier of space-efficient and easily interpretable no-swap regret algorithms.
>
> We hope that our answers, including our commitment to reorganize Section 4, addressed your remaining concerns and if so you would be willing to increase your score. Thank you again for your helpful suggestions!

---

> > ### Comment · Reviewer_b4Uc · 2024-08-11
> >
> > I thank the authors for the response. I have no further questions, and I maintain my positive evaluation.

---

### Author Rebuttal · Authors · 2024-08-07

We thank all the reviewers for their careful and thoughtful reviews of our paper. We respond to each review individually below. (The PDF attached to this global rebuttal contains a figure accompanying a response to a question of Reviewer r1mG).

---

### Decision · Program_Chairs · 2024-09-25

**Decision:**

Accept (poster)

**Comment:**

This paper proves that in symmetric zero-sum games that satisfy some additional assumptions (symmetric initialization), no-swap-regret dynamics will provably converge to Nash equilibrium in the frequent-iterate sense.  This guarantee does not extend to asymmetric games.  It includes additional results that are likely to be significant (e.g., no-swap-regret dynamics cannot be implemented as a time-symmetric function of observed rewards).

Most of the reviewers found the paper's results to be an important contribution to the growing literature on the properties of no-swap-regret algorithms.  One reviewer was concerned that the language of last-iterate convergence was misleadingly strong; the authors proposed using more precise frequent-iterate language in the camera-ready revision, and I encourage them to do so.